# Alteration of Mesenchymal Stem Cells Isolated from Glioblastoma Multiforme under the Influence of Photodynamic Treatment

Kalina Tumangelova-Yuzeir [1],*, Krassimir Minkin [2], Ivan Angelov [3], Ekaterina Ivanova-Todorova [1], Ekaterina Kurteva [1], Georgi Vasilev [1], Jeliazko Arabadjiev [4], Petar Karazapryanov [2], Kaloyan Gabrovski [2], Lidia Zaharieva [5], Tsanislava Genova [5] and Dobroslav Kyurkchiev [1]

1 Laboratory of Clinical Immunology, University Hospital "St. Ivan Rilski", Medical University of Sofia, 1431 Sofia, Bulgaria
2 Clinic of Neurosurgery, University Hospital "St. Ivan Rilski", Medical University of Sofia, 1431 Sofia, Bulgaria
3 Institute of Organic Chemistry with Centre of Phytochemistry, Bulgarian Academy of Sciences, 1113 Sofia, Bulgaria
4 Acibadem City Clinic, University Hospital "Tokuda", 1407 Sofia, Bulgaria
5 Institute of Electronics, Bulgarian Academy of Sciences, 1784 Sofia, Bulgaria
* Correspondence: kullhem000@gmail.com

**Abstract:** The central hypothesis for the development of glioblastoma multiforme (GBM) postulates that the tumor begins its development by transforming neural stem cells into cancer stem cells (CSC). Recently, it has become clear that another kind of stem cell, the mesenchymal stem cell (MSC), plays a role in the tumor stroma. Mesenchymal stem cells, along with their typical markers, can express neural markers and are capable of neural transdifferentiation. From this perspective, it is hypothesized that MSCs can give rise to CSC. In addition, MSCs suppress the immune cells through direct contact and secretory factors. Photodynamic therapy aims to selectively accumulate a photosensitizer in neoplastic cells, forming reactive oxygen species (ROS) upon irradiation, initiating death pathways. In our experiments, MSCs from 15 glioblastomas (GB-MSC) were isolated and cultured. The cells were treated with 5-ALA and irradiated. Flow cytometry and ELISA were used to detect the marker expression and soluble-factor secretion. The MSCs' neural markers, Nestin, Sox2, and glial fibrillary acid protein (GFAP), were down-regulated, but the expression levels of the mesenchymal markers CD73, CD90, and CD105 were retained. The GB-MSCs also reduced their expression of PD-L1 and increased their secretion of PGE2. Our results give us grounds to speculate that the photodynamic impact on GB-MSCs reduces their capacity for neural transdifferentiation.

**Keywords:** glioblastoma multiforme; mesenchymal stem cells; photodynamic therapy; 5-ALA

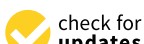



## 1. Introduction

Glioblastoma multiforme (GBM) is a histologically heterogeneous and highly invasive primary brain tumor classified as a grade IV astrocytoma by the World Health Organization (WHO). The central hypothesis for its development postulates that the tumor evolves from the transformation of normal neural stem cells (NSCs) into so-called "cancer stem cells" (CSCs) [1–3]. Along with the role of CSCs, data are accumulating on the role of tumor stroma in GBM development. Under the influence of CSCs, changes in the tumor microenvironment occur that favor tumor progression [4]. In addition to tumor cells, the tumor microenvironment includes lymphatic and blood vessels, chemokines, cytokines, growth factors, extracellular matrix, and non-tumor stromal cells. Mesenchymal stem cells (MSCs) are essential parts of the latter [5,6].

The unravelling the role of GB-MSCs in the tumor stroma began with one of the cells isolated from the GBM cell-culture model. The classical method of culturing is in a serum-free medium containing EGF, where bFGF and tumor spheres containing CSC develop.

In addition, another model has been described. According to this second model, cells are cultured in a serum medium and form fibroblast-like adherent cell cultures (AC) capable of self-renewal. These cells can also express Nestin and GFAP [2], and lack the genetic changes typical of tumor cells and CSCs. Their tumorigenicity is debatable [7]. Some data indicate that ACs have a significantly weaker differentiation capacity in the neuronal direction than CSCs [8,9]. Nakahata et al. first suggested that ACs have MSC properties [10]. Later, Hossain et al. isolated and GBM adherent cell cultures and demonstrated that these met all the International Society for Stem and Gene Therapy (ISCT) criteria for mesenchymal stem cells [11].

Glioblastoma-derived mesenchymal stem cells (GB-MSCs) have a typical MSC morphology and self-renewing ability, express typical MSC markers, CD90, CD73, CD105, and do not express CD45 or CD34 [4,11]. These cells are capable of osteogenic and adipogenic differentiation and often have genetic characteristics similar to those of classical MSCs [1,11]. Our previous research on adherent cell cultures (AC), isolated from GBM, established the simultaneous expression of markers typical of CSCs, such as Nestin, GFAP, Sox2, and CD44, and the mesenchymal markers CD73, CD90, CD105, CD29, CD146, and HLA-I. These cells can differentiate in the osteogenic and adipogenic directions [1]. Therefore, it is clear that, along with CSCs, GBMs contain another population of stem cells in their stroma, namely GB-MSCs [12]. In their experiments, Hossain et al. described three types of GB-MSC, according to their genetic characteristics. The first group has the genetic profile of conventional MSCs, the second group has genetic features similar to those of CSCs, which are thought to be the result of transdifferentiation, and the third group has no genetic features of either MSCs or CSCs, which is thought to result from a process described as "stromal corruption" [11].

A highly significant factor in the tumor stroma is the well-known property of MSCs to modulate the immune response in the direction of immune suppression. This type of action occurs through the direct contact between MSCs and the cells of the immune system through the release of extracellular vesicles and, predominantly, through the paracrine action of cytokines [4,13]. Previous results obtained by our group and foreign groups found that GB-MSCs express PD-L1, as well as secreting numerous cytokines, some of which (TGF-β, CCL-2, PGE2, IL-6, progesterone-induced blocking factor (PIBF)) can induce the suppression of immune cells [1,5]. Under the influence of soluble factors released by GB-MSCs, a decrease in Th17 lymphocytes and an increase in Tregs occurred in our research. In addition, the paracrine activity of GB-MSCs leads to the suppressed expression of CD80 and HLA-DR by monocyte-derived cells [1].

Photodynamic therapy is based on the interaction between a photosensitizer, which predominantly accumulates in pathological tissues, and light with a proper wavelength. The photosensitizer absorbs light with a certain wavelength, which leads to the generation of singlet oxygen and other reactive oxygen species (ROS). The 5-aminolevulinic acid (5-ALA) is classified as a second-generation photosensitizer; it accumulates selectively in the tumor and is characterized by chemical purity. This substance is a precursor of the endogenous photosensitizer protoporphyrin IX (PPIX), with a high ability to generate singlet oxygen. In normal cells, 5-ALA is metabolized to the final product heme and synthesized by PPIX and iron under the influence of enzyme ferrochelatase. It has been proven that applying 5-ALA on malignant gliomas leads to the accumulation of a high quantity of PPIX in tumor cells without leading to the formation of heme molecules. The abnormally high accumulation of PPIX in cancer cells results from a few metabolic alterations. Oxygen-independent glycolysis, known as the Warburg effect, primarily produces ATP in cancer cells. [14]. Additional factors are the up-regulated activity of oligopeptide transporters, which introduces ALA into the cytoplasm of cancer cells, and the down-regulation of other transporters, which provides the efflux of PPIX [15]. Furthermore, the neovascularity of the tumors is irregularly permeable, which allows higher diffusion rates of 5-ALA [16]. Under the influence of light with a wavelength of 630 nm, the ground singlet state (S0) PPIX moves into an excited state (S1). As a result, energy is released, so PPIX passes into

the triplet state (T1). The T1 of PPIX directly transfers energy to the ground triplet state of oxygen molecules in close proximity, leading to the formation of singlet oxygen and transferring electrons to different surrounding molecules, leading, in turn, to the formation of other ROS. A high concentration of ROS causes mitochondrial damage and releases on cytochrome c in the cytosol, which leads to the activation of apoptosome and procaspase 3, prompting apoptosis [17–22]. Our previous results suggested enhanced apoptosis and necrosis in GB-MSCs treated with photodynamic therapy [23].

Reactive oxygen species have a key role in tumor biology, which is far from limited to their destructive effect leading to cell death. It has been established that under their influence, multiple signaling pathways in tumor cells are activated (e.g., MAPK, HIF, NF-kβ), thereby affecting the processes of proliferation, genomic stability, inflammation, and angiogenesis [6]. Under the influence of ROS, angiogenesis is affected by the induction of hypoxia-induced factor (HIF), which leads to the expression of vascular endothelial growth factor (VEGF), while MAPKs and other transcription factors and protein kinases affect the processes of survival, growth, and metabolism [6]. Studies of the influence of ROS on different types of stem cells reveal that ROS affect basic processes related to stem-cell biology: self-renewal, proliferation, differentiation, "stemness", epigenetic regulation, and genomic instability [24]. These effects occur under the influence of a set of sensory and adapter proteins (Wnts, Sox9, HIF-1$\alpha$, JINK, PRAR$\gamma$, FOXO, etc.) [24,25].

In recent years, it has become clear that ROS can act as secondary messengers, influence cell metabolism, and determine cell fate. In healthy cells, they are produced in the mitochondria, most often during the formation of adenosine triphosphate (ATP) [26]. It is assumed that during the cellular resting stage, they have a certain base level, which helps to maintain cellular homeostasis. Moreover, it is well known that in response to sources of stress, such as hypoxia, starvation, infection, or the influence of growth factors, the mitochondrial level of ROS is increased, and they participate in the regulation of signaling pathways in cells through reversible modifications of various molecules. This leads to the activation of important processes, such as adaptation to hypoxia, cell differentiation, proliferation, and autophagy [27].

In the present study, we investigate the changes in surface and intracellular molecules, as well as the secretion of some cytokines by GB-MSCs after photodynamic treatment with 5-ALA (Figure 1).

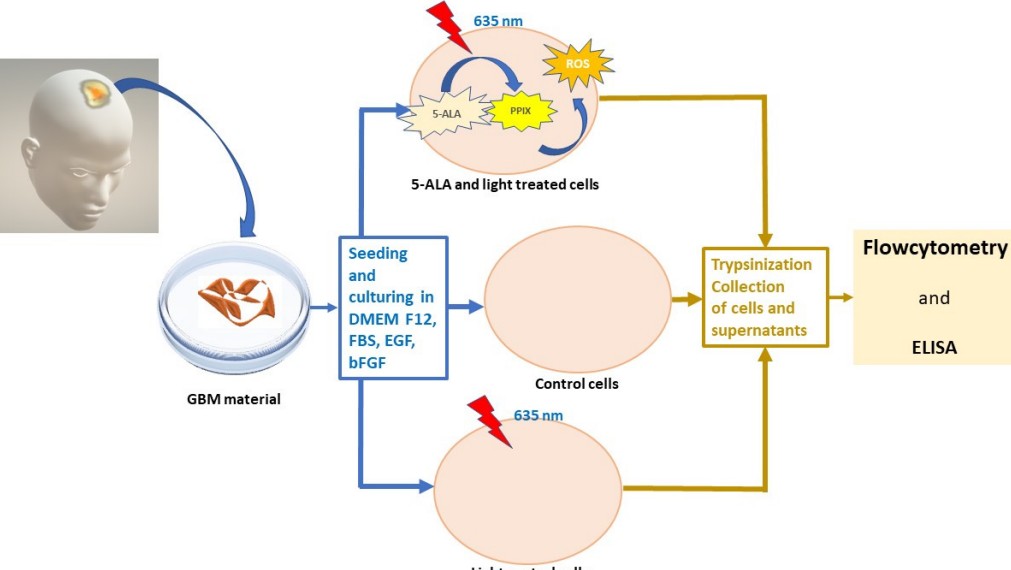

**Figure 1.** Photodynamic treatment of GB-MSCs. Schematic diagram of experimental steps and procedures.

## 2. Materials and Methods

### 2.1. Materials

For this study, brain tissues were collected from 15 patients (seven women and eight men, mean age 58.3 years, range 34–74 years) with a histologically proven diagnosis of GBM (stage IV astrocytic tumor). All patients were newly diagnosed and without any treatment before the surgical intervention. The tissue samples of 2–3 cm$^3$ were obtained during surgical resection of the tumor at the Neurosurgery Clinic at the "St. Ivan Rilski" Hospital in Sofia. All patients included signed informed written consent. The study was conducted in accordance with the study protocol (Medical University of Sofia Ethics Committee Approval No. 1087/14.03.2019), Declaration of Helsinki, and applicable regulatory requirements. Tissue samples were isolated from the so-called "T" zone of the tumor, which is characterized by cell viability and absence of necrosis [28], and then immediately placed in sterile phosphate-buffered saline (PBS), pH 7.4 and delivered to the Laboratory of Clinical Immunology in the hospital.

### 2.2. Methods

#### 2.2.1. Cell Culture

Isolation and culturing of stem cells from the tissue materials were performed in sterile conditions according to a well-established protocol for obtaining GB-MSC cultures using DMEM/F-12 medium (PAN Biotech, Aidenbach, Germany), 10% fetal bovine serum (FBS) (FBS Good, PAN Biotech, Aidenbach, Germany), 10 ng/mL basic fibroblast growth factor bFGF (PAN Biotech, Aidenbach, Germany), and 10 ng/mL epidermal growth factor (EGF) (PAN Biotech, Aidenbach, Germany) [7]. Isolated GBM cells were seeded and cultivated at 37 °C and 5% $CO_2$, and after a period of 7 to 14 days, they formed a monolayer of adherent, fibroblast-like cells. Detailed descriptions of the isolation, culturing, and validation of GB-MSCs regarding their clonogenicity, differentiation, marker expression, and cytokine secretion are described in our previous publication [7].

#### 2.2.2. Photodynamic Treatment

After reaching confluence of cell cultures above 80%, GB-MSCs from every patient were cultured with 5-ALA (ALA HCL, Biotrade, Sofia, Bulgaria, Lot:191260313, concentration of 25 μg/mL). In parallel, we incubated GB-MSCs from every GBM patient without adding a photosensitizer; these were termed control GB-MSCs. Therefore, 15 pairs of control GB-MSCs and GB-MSCs cultured with 5-ALA were prepared. The 5-ALA-treated cells were incubated for 4 h, after which they were irradiated for 20 min. For the irradiation, we used a specially developed LED device that emits light from the visible spectrum with a wavelength of 635 nm, which allows even distribution of the light intensity up to 100 mW/cm$^2$ in the treated area. For the experiment, we used light intensity of 50–60 mW/cm$^2$ and irradiation time of 20 min to obtain a dose of approximately 60 J/cm$^2$ [16]. In six of the experiments, a "light control" was included, in which control cells (without added 5-ALA) were irradiated together with 5-ALA-treated cells. After 5-ALA-treated cells were append and irradiated, treated and control cells were incubated for 24 h at 37 °C and 5% $CO_2$. For the purpose of the subsequent studies, cells were trypsinized (Trypsin 0.05%/EDTA 0.02% in PBS, PAN Biotech, Aidenbach, Germany), centrifuged at $500\times g$ for 10 min, and collected. The supernatants were separated and stored at −80 °C.

#### 2.2.3. Flow Cytometry

After trypsinization, GB-MSCs were brought to single-cell suspension, adjusted to $0.5 \times 10^5$–$1 \times 10^5$ cells per 100 μL, and examined by flow cytometric analysis for the expression of MSC-specific markers: CD73 (Anti-human CD73 PE antibody, eBioscience TM, Thermo Fisher Scientific Inc, Waltham, MA, USA, final concentration 5 μL/100 μL), CD90 (FITC mouse anti-human CD90, BD Biosciences, San Diego, CA, USA) final concentration 5 μL/100 μL), CD105 (PerCP-Cy 5.5 mouse anti-human CD105, BD Biosciences, San

Diego, CA, USA, final concentration 5 µL/100 µL), markers characteristic of CSCs–CD44 (PerCP anti-mouse/human CD44 antibody, BioLegend, Inc., San Diego, CA, USA, final concentration 0.1 µg/100 µL), Sox2 (Alexa Fluor 647 anti-Sox2, BioLegend, Inc., San Diego, CA, USA, final concentration 0.25 µg/100 µL), GFAP (Alexa Fluor 488 anti-GFAP, BioLegend, Inc., San Diego, CA, USA, final concentration 0.1 µg/100 µL), Nestin (PE anti-Nestin, BioLegend, Inc., San Diego, CA, USA, final concentration 5 µL/100 µL), and the inhibitory molecule PD-L1 (APC mouse anti-human CD274, BD Biosciences, San Diego, CA, USA, final concentration 5 µL/100 µL). Since some of the markers were located intracellularly, a kit, Intracellular Fixation and Permeabilization Buffer Set (eBioscience TM, Thermo Fisher Scientific Inc., Waltham, MA, USA), was used for permeabilization of cell membranes following the manufacturer's instructions. Samples were analyzed using a FACSCalibur flow cytometer (Becton Dickinson, San Diego, CA, USA), and BD CellQuest Pro Software (Becton Dickinson, San Diego, CA, USA) was used for the analysis.

### 2.2.4. Immunoenzyme Assay (ELISA)

Following the manufacturer's instructions, we tested 5-ALA-treated and control GB-MSC for IL-1RA (Cat # BMS2080, Invitrogen, Thermo Fisher Scientific, Waltham, MA, USA), TGF-β1 (cat. no. CSB-E0472h, CUSABIO, Houston, TX, USA), CCL5/RANTES (cat. no. abx152923, Abbexa, Cambridge, UK), CCL2/MCP-1 (cat. no. DCP00, R&D System, Inc., Minneapolis, MN, USA), and Prostaglandin E2 (cat. no. KGE004B, R&D System, Inc., Minneapolis, MN, USA).

The entire experiment is schematically described in Figure 1.

### 2.2.5. Statistical Processing

The obtained data were processed and visualized using RStudio (RStudio Team (2020). RStudio: Integrated Development for R. RStudio, PBC, Boston, MA, USA) and GraphPad Prism 8.0 (GraphPad Software, Inc). The normality of the variable was examined using the Shapiro–Wilk test. Due to the small number of experiments ($n = 15$), the Wilcoxon exact test was applied when comparing the effect of photodynamic treatment. Adjustment in compliance with the Benjamini–Hochberg method was applied and q values were calculated. Spearman's correlation analysis was also used to investigate correlations between variables. Any *p* value < 0.05 was accepted as a reliable level of significance.

## 3. Results

### *3.1. Nestin Expression*

We found a significant decrease in the percentage of GB-MSCs with positive Nestin expression after the addition of 5-ALA to the samples, followed by light exposure, in comparison with the control cell cultures, 84 ± 15% (mean ± std. deviation), 90%, 40–97.4% (median, min.–max.) vs. 64 ± 21%, (mean ± std. deviation) 64%, 14.2–92.4% (median, min.–max.), *p* = 0.001, q = 0.0025, one-tailed *p*-value obtained by Wilcoxon signed-rank test, Figure 2.

### *3.2. GFAP Expression*

Regarding the GFAP, two significant differences were also found:

- Under the influence of 5-ALA and light, a significant decrease in the percentage of GFAP-positive GB-MSCs was observed compared to the control GB-MSCs, 91 ± 6.6% (mean ± std. deviation), 93.9%, 80–90.8% (median, min.–max.) vs. 80.6 ± 15% (mean ± std. deviation), 84.9%, 48–97.5% (median, min.–max.), *p* = 0.001, q = 0.0025, Figure 3. The decrease in the percentage of GFAP-positive GB-MSC cells was positively correlated with the diminishment in the percentage of Nestin- positive GB-MSC, s Spearman's rho = 0.614, *p* = 0.015.
- The mean fluorescent intensity (MFI) of the GFAP expression also diminished after 5-ALA and light exposure—367 ± 364 (mean ± std. deviation), 242, 63–1562 (median,

min.–max.) in the control GB-MSCx vs. 287 ± 312 (mean ± std. deviation), 191, 106–136 (median, min.–max.) in treated cells, *p* = 0.005, q = 0.0083, Figure 4.

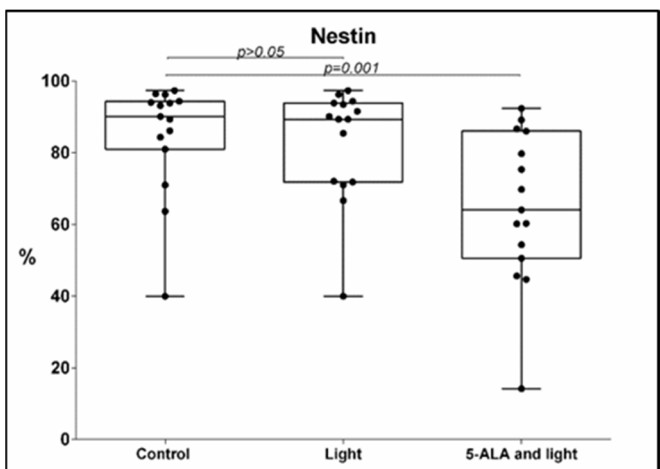

**Figure 2.** Boxplot showing the change in the percentage of Nestin-positive GB-MSCs across paired samples of control cell cultures, cell cultures treated with light only, and cell cultures exposed to 5-ALA and light. Nestin expression was evaluated using multi-color flow cytometry, and every dot on the box plot represents the percentage values of all Nestin-positive GBM-MSCs in that particular cell culture.

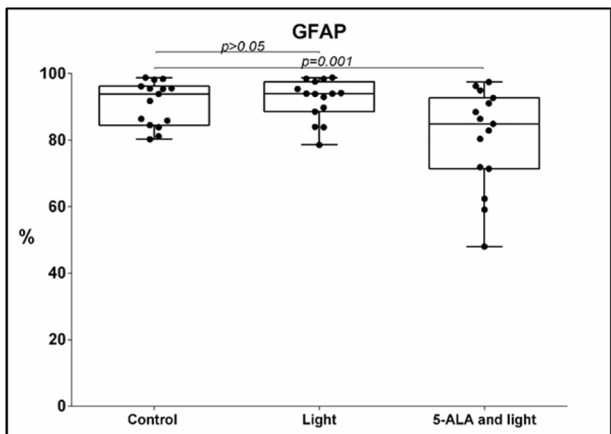

**Figure 3.** Boxplot showing the change in the percentage of GFAP-positive GB-MSCs across paired samples of control cell cultures, cell cultures treated with light only, and cell cultures exposed to 5-ALA and light. The GFAP expression was evaluated using multi-color flow cytometry, and every dot on the box plot represents the percentage values of all GFAP-positive GBM-MSCs in that particular cell culture.

*3.3. Sox2 Expression*

The GB-MSCs under the influence of the 5-ALA and light exposure had significantly lower expressions of Sox2 compared to the control GB-MSCs, a median MFI of 628 (24–2794) (median, min.–max.) vs. 834 (24–4698) (median, min.–max.), *p* = 0.015, q = 0.016. The Sox2 MFI was positively correlated with the GFAP MFI. Spearman's rho = 0.611, *p* = 0.016, Figure 5.

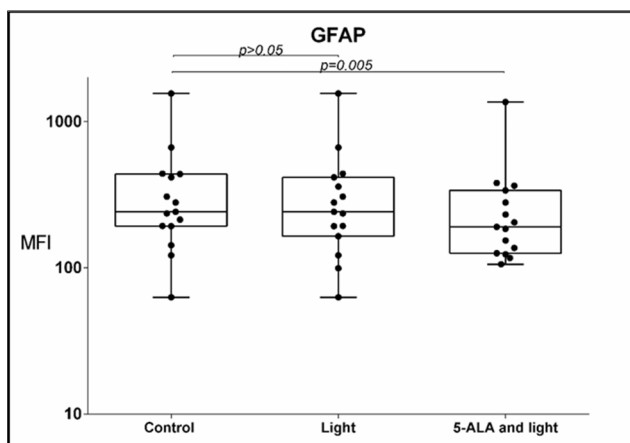

**Figure 4.** Boxplot showing the change in MFI levels of GFAP expression by GB-MSCs across paired samples of control cell cultures, cell cultures treated with light only, and cell cultures exposed to 5-ALA and light. The MFI GFAP expression was evaluated using multi-color flow cytometry, and every dot on the box plot represents the values of all GFAP-MFIs in that particular cell culture.

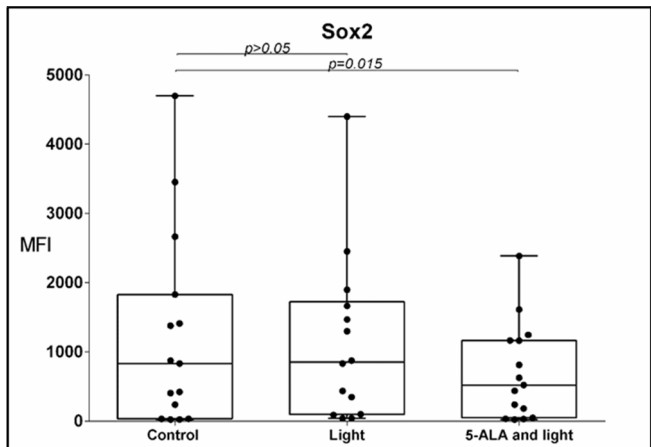

**Figure 5.** Boxplot showing the change in the percentage of Sox2-positive GB-MSCs across paired samples of control cell cultures, cell cultures treated with light only, and cell cultures exposed to 5-ALA and light. The Sox2 expression was evaluated using multi-color flow cytometry, and every dot on the box plot represents the percentage values of all Sox2-positive GBM-MSCs in that particular cell culture.

*3.4. PD-L1 Expression*

We found a decrease in the percentage of PD-L1 positive GB-MSCs treated with 5-ALA and light, $89 \pm 8.1\%$ (mean $\pm$ std. deviation), 90.3%, 79.3–98.5% (median, min.–max.), compared to the control GB-MSCs, with $93 \pm 6.1\%$ (mean $\pm$ std. deviation), 94.3%, 82.6–99.2% (median, min.–max.), $p = 0.016$, q = 0.016, Figure 6.

*3.5. Expression of CD105, CD73, CD90, and CD44*

No significant variation in the expression of CD105, CD73, CD90, or CD44 was observed.

*3.6. Changes in Levels of TGF-β, CCL-2, CCL-5, IL-1RA, and PGE2*

Exposure to 5-ALA and light did not result in significant alterations in the levels of TGF-β, CCL-2, CCL-5, and IL-1RA, $p > 0.05$. However, a significant increase in the levels of PGE2 was observed. The median of concentration was 900 pg/mL and min.–max levels were 194–3041 in the control GB-MSCs, compared with 2200 pg/mL (557–4678) in the 5-ALA-and-light-treated cultures, Figure 7.

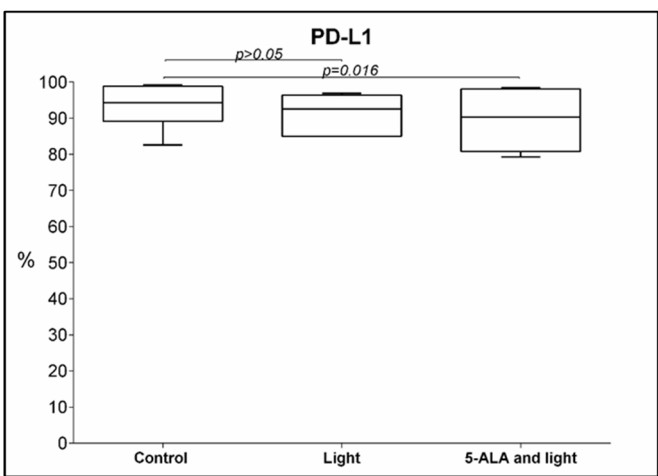

**Figure 6.** Boxplot showing the change in the percentage of PD-L1-positive GB-MSCs across paired samples of control cell cultures, cell cultures treated with light only, and cell cultures exposed to 5-ALA and light. The PD-L1 expression was evaluated using multi-color flow cytometry, and every dot on the box plot represents the percentage values of all the PD-L1 positive GBM-MSCs in that particular cell culture.

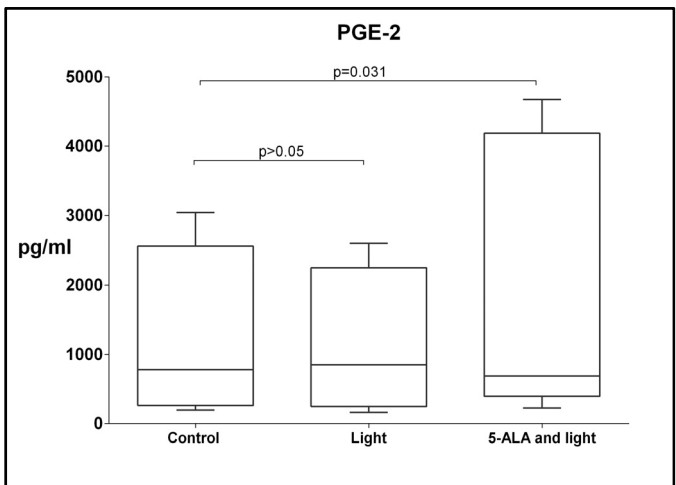

**Figure 7.** Boxplot showing the change in the levels of PGE2 across paired samples of control cell cultures, cell cultures treated with light only, and cell cultures exposed to 5-ALA and light. The effect of the PGE2 secretion on the protein levels was evaluated using ELISA Kit for human PGE2. Every dot on the box plot represents the levels of PGE2 in pg/mL produced by GBM-MSCs in that particular cell culture.

For the visualization of the alterations in the markers of the GB-MSCs under the influence of 5-ALA and light exposure, a radar plot was generated using the Z scores of the percentages of Nestin, GFAP, Sox2, and PD-L1, as well as their corresponding MFI values. The radar plot shows, simultaneously, the changes between the control GB-MSCs and GB-MSCs exposed to 5-ALA and light (Figure 8).

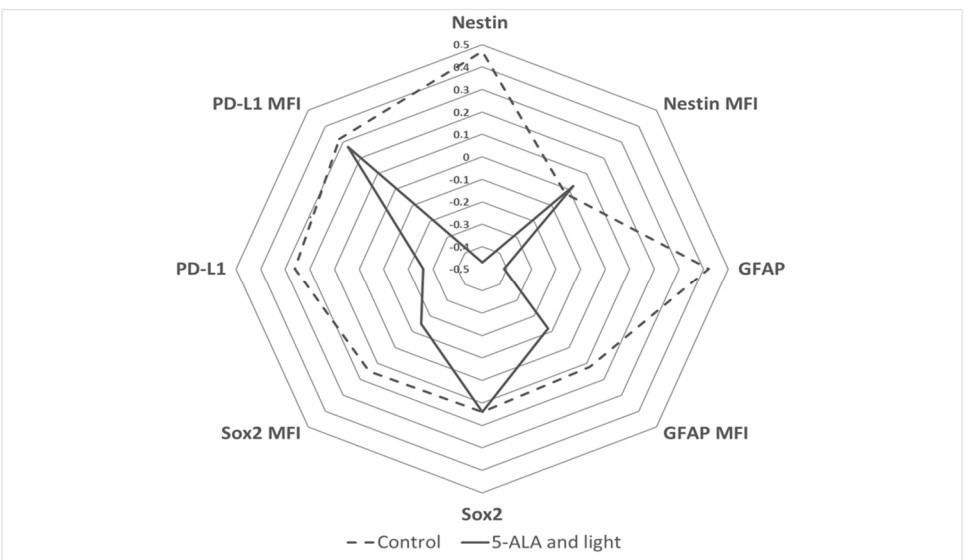

**Figure 8.** Radar plot summarizing the previous results of the observed percentages of Nestin, GFAP, PD-L1, and Sox2-positive GB-MSCs and the mean fluorescence intensity (MFI) of expression of Nestin, GFAP, PD-L1, and Sox2 markers at single cellular levels in GB-MSCs. Features' values are mean-centered and divided by one standard deviation for better visual generalization. The punctured line depicts the feature values among control GB-MSCs, and the bold line depicts GB-MSCs exposed to 5-ALA and light. The radar plot demonstrates the decreased percentages of Nestin, GFAP, and PD-L1 expression by GB-MSCs treated with 5-ALA and light and the lack of change in the Nestin, GFAP, and PD-L1 MFI, with no alteration in the percentage and MFI of Sox2 expression.

## 4. Discussion

There are different hypotheses concerning the role of GB-MSCs in the tumor stroma. Mesenchymal stem cells are known to have tumor tropism and are able to migrate to ischemic and injured tissues following a chemokine gradient [1,4,5]. Multiple data indicate that MSCs accumulate in gliomas, occupying perivascular niches [4]. Another possibility is that GB-MSCs are products of CSC transdifferentiation. It is known that CSCs can transdifferentiate from a neural to a mesenchymal direction, and the factors involved in this process—cEBP13, STAT-4, and MLK4—have been described [29,30]. Both GB-MSCs with genetic profiles of MSCs and GB-MSCs with genetic profiles of CSCs are found in the GBM tumor environment [12,31]. As is well known, the WHO divides GBMs into four subtypes, one of which is designated as "mesenchymal". This type is more aggressive and resistant to therapy, and it has been found that in this variant, CSCs express mesenchymal markers, such as CD90 [4].

A third possibility is that GB-MSCs in the tumor environment are MSCs that have undergone a partial process of neural transdifferentiation. Mesenchymal stem cells were found to possess 12 genes and 11 transcription factors associated with neural tissue. In other words, they have a genetic predisposition for transdifferentiation in a neural direction. This includes the genes for Nestin and GFAP, and it has been found MSCs can express them. This happens while, at the same time, MSCs retain the expression of their typical markers and their ability to differentiate into mesenchymal lineages [12,32,33]. At least in vitro transdifferentiation in a neural direction has been repeatedly demonstrated for MSCs of different origins [7,12]. It has been suggested that brain MSCs have a more remarkable ability to transdifferentiate in a neural direction than MSCs isolated from other sources [12]. Some authors assign a central role to MSCs in tumor development, considering that mutations primarily affect perivascular MSCs, which migrate into the brain parenchyma. There, under the influence of the environment, GB-MSCs transdifferentiate into CSCs. The proliferation of CSCs leads to hypoxia and neoangiogenesis. As a result of these altered conditions, CSCs again transdifferentiate into GB-MSCs [30]. There is also

skepticism towards the idea of MSC transdifferentiation, suggesting that fusion between MSCs and neurons rather than true transdifferentiation is observed [12,34].

A fourth possibility for GB-MSCs in the tumor microenvironment is that these cells represent conventional MSCs undergoing a process known as "stromal corruption"—the ability of a tumor to influence its stromal elements, manipulating them in order to sustain its development [35].

Regardless of their origin, as parts of tumor stroma, GB-MSCs exert an intense influence on CSCs. They stimulate their proliferation and tumorigenicity [4,11] and participate in GBM angiogenesis. This is evidenced by the establishment of CD105+GB-MSC around the tumor arterioles [11,36].

Reactive oxygen species accompany the formation and progression of tumors, as their increased level in cancer cells leads to damage to macromolecules and to mutations [37]. Since malignancies develop under hypoxic conditions, hypoxia-induced ROS support tumor-cell activation and survival, similar to stress-induced activation in healthy cells. Recently, hypoxia-induced ROS have also been shown to be major factors in the epithelial-to-mesenchymal transition (EMT) process characteristic of invading and metastasizing CSCs [38].

On the other hand, as previously mentioned, the overproduction of ROS induces the destruction of DNA and other macromolecules in cells, leading to cell death [39]. It is hypothesized that variations in the ROS lead to different effects on cell fate. The baseline level of mitochondrial ROS is involved in maintaining the homeostasis of cells in a resting state. At the same time, their increase to a certain threshold under stress leads to cell activation and survival, and their uncontrolled overproduction causes the irreversible destruction of macromolecules and cell structures and triggers cell-death pro-grams [26,38]. This type of uncontrolled ROS formation, leading to cell death, is observed in photodynamic therapy (PDT). Photodynamic treatment is a method used in neurosurgery to establish resection margins for GBM based on emitted light. Along with light emission, it is well known that PDT leads to the generation of ROS. The results we obtained in a previous article show that PDT treatment leads to increased apoptosis and necrosis in GB-MSCs [40]. ROS formation is the most probable cause of cell death observed after PDT treatment. Because of the clearly established elevation of ROS after 5-ALA and light treatment, we can speculate that the alteration of investigated markers is a consequence of excess ROS.

Our results show that, under the action of photodynamic treatment, GB-MSCs decrease their expression of Nestin, Sox2, and GFAP, with no significant gender-related differences in any of the parameters studied (Figure 9). One of the essential issues in our research was whether, under PDT, our cells would stay alive. The method we used to determine the expression of cell markers (flow cytometry) allows vital cells to be detected by their main characteristics: size and granularity. The method detects the percentage of positive cells (for a given marker) from the cellular population. Furthermore, it displays the mean expression intensity of the marker of the cell population (MFI). Therefore, by using this method, we receive information on variations in markers' expression associated with changes in their cell metabolism.

Nestin is primarily a cytoplasmic microfilament protein correlating with the stemness property. It is a plastic cytoskeletal protein whose expression is influenced by the functional state of the cell [3]. Nestin has a role in the organization of the cytoskeleton, cell signaling, and metabolism. Cancer stem cells express Nestin and, upon differentiation, this expression decreases [2]. There are many data demonstrating that the expression of Nestin is affected by the oxygen level, as well of the ROS level. In a study of Muller glial cells and astrocytes isolated from rat retinas, it was found that treatment with 80% O2 for 2 h resulted in a strong decrease in Nestin expression [3]. Furthermore, under the influence of ROS, neural stem cells decrease their expression of Nestin and Sox2, which is related to other changes in their effects on the processes of apoptosis and self-renewal [41,42]. In MSCs specifically, antioxidants have been shown to enhance their transdifferentiation in a neural direction, a process that is accompanied by the increased expression of Nestin and the decreased

expression of Notch-1 [43]. Our results show that, under PDT, GB-MSCs decrease their expression of Nestin. Therefore, the influence of ROS on Nestin, leading to its reduction, could indicate a limitation in the process of pro-neural transdifferentiation. Alternatively, the reduced expression of Nestin could be related to a process of autophagy stimulated by ROS activity [6].

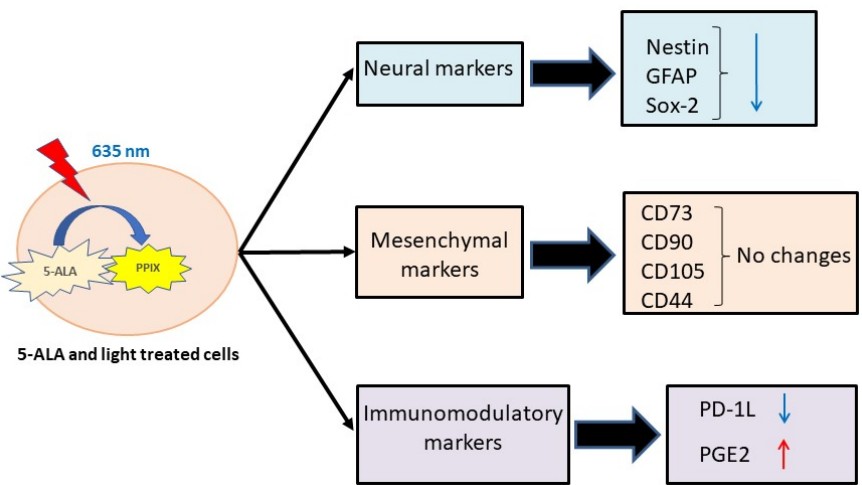

**Figure 9.** The summary of the results obtained for the effect of PDT on GB-MSCs. The PDT leads to the down-regulation of markers associated predominantly with neural differentiation (Nestin, Sox-2, and GFAP). The markers associated predominantly with the mesenchymal phenotype (CD73, CD90, CD105, and CD44) retain their expression. According to the substances engaged with immunomodulation, PD-1L decreases its expression, but PGE2 demonstrates elevated secretion levels.

Glial fibrillary acid protein is a marker of astrocyte differentiation. In CSCs, it is not usually expressed, but its co-expression with Nestin can be observed [2]. The ACs express GFAP alone or together with Nestin [7,8,44,45]. The co-expression of GFAP and Nestin in GB-MSCs was described by our team in our previous publication [1]. Studies of astrocytes have shown that, under the action of ROS, they are activated and increase their expression of GFAP [46–48]. Our results for the GB-MSCs show the opposite process: reduced expression under the influence of ROS. To our knowledge, this is the first report on the effect of ROS on stem cells in terms of GFAP expression. Although our data are opposite to those obtained for astrocytes, stem cells may behave differently from differentiated cells, especially regarding the impact of ROS. There is also a fundamental difference between ROS derived from oxidative stress or hypoxia and ROS derived from photodynamic therapy. As discussed above, ROS obtained differently could have different effects. However, further research is needed on this question.

The Sox family of transcription factors are important regulators involved in the development, regeneration, and homeostasis of various tissues in the body, as well as in processes of cell-fate reprogramming [49]. One of the main representatives of this family is sex-determining region Y-box 2 (Sox2), which is a critical factor supporting the self-renewal and pluripotency of embryonic stem cells. It is also expressed by stem cells in some tissues of the adult organism, such as NSCs in neurogenic regions of the subventricular zone of the lateral ventricle, the subgranular zone of the hippocampus, and the ependyma of the adult central canal [49]. Furthermore, Sox2 is actively involved in the processes of CSC formation, proliferation, EMT, migration, invasion, metastasis, resistance to apoptosis, and therapy [50]. According to the main theory of GBM origin, CSCs result from NSC transformation and GBM tissue sections are expected to show increased levels of Sox2. In addition, Sox2 has been shown to have a key role in GBM progression and recurrence, because its silencing in GBM CSCs leads to a drastic reduction in their capacity for invasion, proliferation, and tumorigenicity [51]. Mesenchymal stem cells also express Sox2 to maintain stemness, proliferation, and proper differentiation; Sox2 expression is down-regulated

during their differentiation [52]. There is evidence that hypoxia-induced ROS up-regulate Sox2 expression, both in malignant cells and in healthy keratinocytes [38,53], in response to oxidative stress and in an effort to ensure cell survival [38]. However, other studies indicate that, under the action of a high concentration of ROS, the expression of this transcription factor decreases, suggesting that ROS above a certain concentration can have the opposite effect and suppress the self-renewal of stem cells [54]. In agreement with the latter, our results also indicate a reduction in Sox2 in GB-MSCs under the action of photodynamic treatment. Photodynamic therapy and the accumulation of ROS in malignant cells in this type of treatment can be used to reduce the self-renewal, tumorigenicity, and invasive abilities of these cells. The study by Gholizadeh et al. supports the idea of the benefits of PDT by showing that the application of the photosensitizer zinc phthalocyanine on a colorectal carcinoma cell line reduced the expression of Sox2. The authors associated this result with the inhibited cell clonogenicity, and also with reduced tumor migration and increased autophagy [55].

Programmed death-1 receptor (PD-1) and its corresponding ligand, PD-L1, forms a major pathway that helps to regulate the immune response, participating in the maintenance of tolerance and protection against autoimmune reactions [55]. Programmed death-1 receptor is mainly expressed on B and T lymphocytes, but can also be expressed by natural killer cells, monocytes, and dendritic cells [56]. In contrast, PD-L1 is widely expressed, both by cells of the immune system and by other cell types (vascular endothelium cells, liver nonparenchymal cells, mesenchymal stem cells, pancreatic islets, astrocytes, neurons, and keratinocytes), which represent the exceptional importance of these molecules for the regulation of the immune system [57]. This major pathway of immune-system regulation, however, is actively used by various tumors. The expression of PD-L1 is one of the main strategies used by tumor cells for immune-response evasion, while PD-1 is highly expressed by tumor-infiltrating lymphocytes [58]. New findings suggest that PD-L1 has another role in the tumor microenvironment and acts as a pro-tumorigenic factor, activating the ability of tumor cells to proliferate and survive [59]. Evidence in this direction was also provided by Qui et al., who showed that the overexpression of PD-L1 in GBM promotes EMT and tumor invasion through RAS/ERK/EMT activation [60]. In addition, studies on hepatocellular carcinoma have shown that Sox2 can directly bind to the promoter region of PD-L1 and activate its expression, which, the authors of one study suggest, leads to tumor-cell proliferation [61]. In our data, the observed high co-expression of Sox2 and PD-L1 could be explained by the same mechanism.

As already mentioned, PD-L1 is expressed by non-immune cells, including MSCs. It is known that, induced by INFγ, MSCs increase the expression of this molecule and also succeed in inhibiting both the proliferation of T cells and their effector functions [62]. Reactive oxygen species have been reported to have a significant but divergent effect on PD-L1 expression [63]. It is most likely that the amount of ROS produced, as well as the mechanisms through which they are induced, are the reason for the observed differences in the effect. Hypoxia, which leads to a modest increase in ROS and cellular activation, rapidly and selectively up-regulates PD-L1 expression in some solid tumors in a manner dependent on the transcription factor HIF1$\alpha$ [64]. The hypoxia-induced up-regulation of PD-L1 in tumors is accompanied by the infiltration of myeloid-derived suppressor cells (MDSC), Tregs, and tumor-associated macrophages (TAM), which further stimulate the expression of this molecule by tumor cells [65].

The influence of many chemotherapeutics used in the treatment of various cancers on the expression of PD-L1 is also known. All of them lead to an increase in oxygen radicals in cells, but have different effects on PD-L1, depending on the mechanism used. Medicines such as auranofin, arsenic, trioxide trifluoperazine, and disulfiram increased the expression of PD-L1 [63], while ethaselen, butaselen, and chaetocin, which induce ROS accumulation due to the inhibition of the antioxidant enzyme, TrxR1, are associated with the down-regulation of PD-L1. It is interesting to note the influence of the photosensitizer, verteporfin, which significantly reduced PD-L1 expression in head-and-neck squamous

carcinoma cells [66]. Our data show similar results in GB-MSCs, with a reduction in the PD-L1 expression after the 5-ALA treatment.

We could not find an alteration in the expression of the typical MSC markers CD105, CD73, CD90, and CD44 [67] under PDT. We consider that under PDT, GB-MSCs show a reduced expression of markers associated with a neural phenotype, which may be associated with a transdifferentiation process in a neural direction, but with not markers typical of the mesenchymal phenotype. At this stage of our work, we have no explanation for this "selective" effect.

Of the five cytokines and soluble factors examined, we reported a statistically significant difference only in the levels of PGE2 under the influence of photodynamic therapy. The PGE2 is one of the products of the conversion of arachidonic acid into PGH2 under the influence of Cox-1/Cox-2 enzymes [68]. The increase in the concentration of ROS in the cell cytosol above a certain threshold level under the influence of photodynamic therapy is an activation stimulus for Cox-1, which is expressed constitutively, while Cox-2 is dependent on proinflammatory stimuli or endoplasmic stress [69,70]. The ROS levels also activate the next enzymatic step, namely the conversion of PGH2 to PGE2 by the membrane-bound enzyme mPGEs-1/2. If we speculate and link the two well-known mechanisms from cell biology, namely that 5-ALA and light are sources of ROS in mitochondria and that ROS influences prostanoid synthesis, we can explain the effect we observed. The PGE2 itself has a well-defined immunosuppressive effect [13]. There is evidence that PDT causes the increased expression of PGE2 and Cox-2 in tumor cells and immune cells, that PGE2 released by dying cells can function as a damage-associated molecular-pattern inhibitor, and that the blocking of PGE2 improves the efficacy of PDT [71,72]. These effects of PGE2 secretion can be interpreted as the induction of a survival mechanism. No similar experimental set-ups examining the PGE2 levels in GB-MSCs are available in the literature, nor have any studies been conducted on photodynamic-therapy ROS formation in MSCs isolated from GBM stroma. In addition, it is important to discuss the fact that other prostanoids, such as PGI2, TXA2, and PGD2, are also probably synthesized during the PDT. In other words, the discussion of the biological "consequences" of secretory factors in the tumor environment should not be related only to PGE2 [13].

## 5. Conclusions

In conclusion, our results show that photodynamic therapy affects GB-MSCs localized in the tumor stroma of GBM. They maintain their mesenchymal phenotype, but lose important markers that define them in the stem and proneural directions. Considering the important roles of these cells as factors in the suppression of the immune response and as a population capable of transdifferentiation, this effect could be relevant for the development of therapeutic approaches to the treatment of GBM.

## 6. Limitations

Our study was limited by the fact that only the change in the neural stem markers was used to analyze how the GB-MSCs were affected by the 5-ALA and the irradiation. Their ability to self-renew and differentiate should also be included in the comprehensive description of GB-MSC alterations. Our present research focuses on how 5-ALA and light affect these characteristics. However, we tested the effect of light only, but our work did not investigate the independent effect of 5-ALA as a control. The reason for this is that according to the research data, 5-ALA without irradiation does not elicit alterations in cell cultures [73,74]. Moreover, our previous studies [23,40] investigated the independent effect of 5-ALA on the processes of apoptosis and necrosis in GB-MSCs. We found no changes compared to untreated GB-MSC controls. The major limitation of our work is the higher dose (60 J/cm$^2$) we used. According to the research data, doses up to 50 J/cm$^2$ are usually used. The use of the dose of 60 J/cm$^2$ is probably not possible to use directly in clinical practice for PDT in patients with glioblastoma. However, the effect of PDT on MSCs could

be a significant focus for future research on therapies focused on MSCs— vital parts of the tumor stroma.

**Author Contributions:** Conceptualization, D.K. and K.M.; methodology, K.T.-Y.; software, G.V.; validation, E.I.-T., K.G. and I.A.; formal analysis, E.K.; investigation, P.K., L.Z. and T.G.; data curation, K.T.-Y. and P.K.; writing—original draft preparation, K.T.-Y., E.K. and G.V.; writing—review and editing, E.I.-T., J.A. and D.K.; visualization, K.T.-Y. and G.V.; supervision, D.K., K.M. and I.A.; project administration, P.K.; funding acquisition, J.A. All authors have read and agreed to the published version of the manuscript.

**Funding:** This research was funded by National Science Fund, Ministry of Education and Science, Bulgaria, grant number КП-06-Н23/8-18.12.2018г. The APC was funded by Bulgarian Scientific Society of Immuno-oncology (BSSI).

**Institutional Review Board Statement:** The study was conducted in accordance with the Declaration of Helsinki, and approved by the Ethics Committee of Medical University of Sofia, Ethics Committee Approval (protocol no. 1087/14.03.2019).

**Informed Consent Statement:** Informed consent was obtained from all subjects involved in the study.

**Data Availability Statement:** The data presented in this study are available upon request from the corresponding author. The data are not publicly available due to local regulations and hospital restrictions.

**Acknowledgments:** We would like to express our sincere gratitude to and deep respect for our beloved colleague and co-author, Ekaterina Borisova. She passed away in April of 2021. Ekaterina Borisova was a dedicated professor at the Institute of Electronics, Bulgarian Academy of Sciences, with a passion for research pertaining to biophotonics in medicine.

**Conflicts of Interest:** The authors declare no conflict of interest. The funders had no role in the design of the study; in the collection, analyses, or interpretation of data; in the writing of the manuscript; or in the decision to publish the results.

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
