# Peer review of "Alteration of Mesenchymal Stem Cells Isolated from Glioblastoma Multiforme under the Influence of Photodynamic Treatment"

_cimb, doi:10.3390/cimb45030169_

Round 1

Reviewer 1 Report

In the Manuscript, effects of photodynamic therapy on the metabolism of glioblastoma cells are demonstrated and discussed in detail. The subject of the study seems to be urgent enough. Understanding the biochemical processes within glioblastoma and the influence of reactive oxygen species on cells' growth and proliferation contribute significantly to the development of effective approaches to treatment  brain tumors.

The study is performed on a high scientific level, results are clearly presented and exhaustively explained. However, there are still some questions to be solved prior to publication the paper.

1. In Introduction, several recent references on the selective uptake of 5-ALA by glioblastoma cells should be added (e.g. Cancers 2021, 13, 580; Transl. Biophotonics 2020, 2, e201900022, Lasers Durg. Med. 2018, 50, 399-419 or so on...).

2. P. 4, line 53: "370 C" should be "37 oC".

3. There were no significant variation in expression of CD105, CD73, CD90, CD44 in control group and in the group treated with PDT. However, this fact is not discussed in "Discussion" part at all. It's necessary to explain the reason why these markers were chosen and why PDT has not affected them.

4. Radar plot (Fig. 7) without explanations on the relationship between Nestin and Nestin MFI, GFAP and GFAP MFI and so on is not informative. The scale labels are hardly visible; the correlation between Fig. 7 and previous figures (1-6) is somewhat unclear. 

5. Cell viability after exposure by PDT should be discussed in more detail. Does the method allow decreasing the number of cells or just affects the cell metabolism? The issue of cell viability is important from the standpoint of practical valuability of PDT.

I believe that minor improvements would make possible to publish the Manuscript in the Current Issues in Molecular Biology journal.

Reviewer 2 Report

It is will known that incubation of malignant cell types with ALA can result in circumvention of a feedback pathway, resulting in formation of protoporphyrin IX. This is  a fluorescent molecule which, upon irradiation, can produce cytotoxic reactive oxygen species. Subsequent irradiation can result in photodamage in affected tissues, usually via an apoptotic response. High PDT doses can also result in necrosis. In the context of glioma, the resulting fluorescence has been used to aid in tumor localization. 

The authors appear to have missed most of the major points. Accumulation of PpIX does not result from variation in iron or ferrochelatase levels, but from bypassing a feedback loop that controls PpIX formation. Line 95 claims that 650-800 nm light is required, but the major long wavelength Q band is at 630 nm. The major effect of reactive oxygen species (ROS) formation is mitochondrial damage leading to the loss of cytochrome c into the cytoplasm, a trigger for apoptosis. High light doses may also lead to necrosis. It is true that ROS can have many effects but in the context of PDT, the major effect is cell death via apoptosis. There can also be an element of vascular shutdown which, in vivo, can also affect survival of malignant cell types.

In the preset report, a 60 J/sq cm light dose is orders of magnitude above what is usually required to kill photosensitized cells. A collection of protein levels is examined 24 h after irradiation. Some minor alterations were detected, but it is important to remember that this will be a population of mainly dead and dying cells and apoptotic cell fragments. The degree of magnification in Fig. 8 is much to low to reveal any pertinent information.   

A Journal of Physics report (Ref 33) also shows that incubation with ALA results in PpIX formation. A collection of undefined procedures were used to evaluate appearance of apoptosis and necrosis. 

The Abstract proposes that photodamage might reduce the capacity of mesenchymal stem cells for differentiation. This may be true, but the major effect is going to be cell death, which will certainly limit differentiation. There are no studies indicating survival of cells after irradiation. Most likely, at the light dose used, there will be no proliferating cells.

Reviewer 3 Report

The paper is interesting, however, some errors need to be corrected before it may be accepted for publication: 

1. No information about antibodies concentrations in the working solution is provided - please add.

2. I think that adding a brief description of data capturing methods below the graphs would help the readers a lot. 

3. The authors used quite high irradiation energy 60J/cm2 - why so? And if it was possible please add more about that into discussion. 

4. In your experiment, the authors have not included the control with 5-ALA. The drug itself can still induce some changes.

5. I think that fig. 8 should be presented in the end of introduction section to make the reader familiar with the study setup. 

6. Section 3.7 can be included in the discussion if no differences were found. 

7. Please add more clinical characteristics of the patients - the disease stage, if it was subjected to chemotherapy or any other kind of treatment, maybe some early followup?

8. Maybe adding a schematic representation of the results in the end of the paper would help to better understand the idea of the study? 

Round 2

Reviewer 2 Report

In this revision, several issues remain.  Abstract lines 23-24 need to be modified. Photodynamic therapy aims to selectively accumulate a photosensitizer  <in neoplastic cells> forming reactive oxygen species (ROS) <upon irradiation> <which initiates death pathways>.  Line 87: singlet oxygen IS a reactive oxygen species. Line 115: change ‘relies on’ to ‘releases’ (cytochrome c into the cytosol). Line 118: what ‘essential metabolites’??? It is not clear that formation of protoporphyrin will cause any vascular effects (line 123). These usually arise from systemic administration of photosensitizers that can photosensitize the tumor vasculature since they will be in the circulation. Protoporphyrin usually does not enter the circulation.

Images of cells in Fig. 1 are at to low a level of magnification to reveal anything. These need to be phase-contrast images at 100X or greater magnification to detect any alterations. This is also true for other figure where there is a depiction of control/treated cells.  Strictly speaking, ALA is not a photosensitizer (187), it is a precursor of a photosensitizer. As pointed out before, an in vitro light dose of 60 J/sq cm is a very high value. 

Irradiation of cells containing protoporphyrin can lead to cell death via apoptosis and (if high light doses are involved) necrosis. This can be assessed via clonogenic assays. After irradiation, cells are incubated for 24 hours, then analyzed for several markers. There is no estimate of cell death, but with this high light doses, the population of non-viable cells could be greater than 90%. There are clearly effects on expression of several proteins (Figs. 2-6) and some are unaffected (section 3.5). 

In the author response, it is indicted that: ‘We detected by flow cytometry that the percentage of necrotic cells is 1.10±0.96%, which is more than one order of magnitude lower compared with the percentage of apoptotic cells 30.03±10.16%. These results are included in another paper, which is accepted for publishing.’  This means that a 60 J/sq cm light dose is only killing ~40% of the cell population. Clonogenic assays are needed to conform this suggestion. Assuming that this is true, such a loss could be replaced by less than one cell division. How can ALA-mediated PDT be proposed for control of glioma if it is this ineffective?   

The authors do ask whether cells can survive photodamage (line 409). Size and granularity (line 411) are proposed as markers for viability but this can only be determined by clonogenic assays. What might be interesting is to examine the viable population after irradiation of photosensitized cells, and see what cell types are present. I suspect that after a light dose of 60 J/sq cm, fewer than 60% of the population will be viable.  

Major concerns with this report are therefore [1] the very high light dose used, [2] the lack of clonogenic assays which would reveal the surviving population of viable cells, [3] the relatively small effects observed in some of the figures, [4] whether the effects observed are simply the result of a variable population of photodamaged cell and photodamaged cellular components. Since the protoporphyrin derived from ALA will concentrate in mitochondria and their outer membranes, it is likely that if effects on mitochondrial structural proteins were examined, changes would also be detected. Neural transdifferentiation may be reduced after PDT, but this could be because dead cells do not transdifferentiate. 

Reviewer 3 Report

The paper was improved however, Fig.7 still has a different style than the other figures before it. Please change that. Also, the comment to my previous concern: "In your experiment, the authors have not included the control with 5-ALA. The drug itself can still induce some changes." should be included in the paper - maybe discussion?

Author Response

Please seethe attachment.

Round 3

Reviewer 2 Report

This Manuscript contains considerable extraneous material. The authors need to concentrate on what is new, do some proofreading,  and not try to recapitulate the entire field of PDT. Readers of this journal will be familiar with the subject. The authors indicate that two types of stem cells are to be dealt with if gliomas are to be successfully eradicated. There is a proposal that MSC cells can ‘give rise to’ SCS. It is being proposed that this transition can be inhibited by PDT based on administration of ALA.      

There remain issues with the presentation. A few minor spelling errors persist, e.g., line 86 wavelength; line 87: replace ‘another’ with ‘other’. ALA does not accumulate selectively in tumor. ALA is converted to PpIX in tissues lacking a feedback mechanism. Photosensitizers do not accumulate in rapidly dividing cells  (line 121). Most tend to accumulate in liver, spleen, kidney and pituitary but are not irradiated and are therefore harmless. 

As pointed out before, magnification of cells in images is too low to permit drawing any conclusions. The light dose (60 J/sq cm) is high for in vitro studies. The pertinent question is whether the typical light dose provided during treatment based on ALA administration will be sufficient to elicit any of the effects indicated in this report. If a very high PDT dose is required to see any of the effects described in this report, it is unlikely that any of these effects will occur during clinical PDT involving ALA. 

It appears that CSC are quite resistant to photodynamic effects. Propagation is presumably not the issue, with the main effect presumably being inhibiting the transition to CSCs. The authors indicate: ‘The test (for viability) we used before is based on Annexin V and Propidium iodide expression. This test is a primarily accepted tool for detecting early apoptosis, late apoptosis and necrosis. This test found that 30.03% of GB-MSCs are in apoptosis, and only 1.1% are in necrosis’. This is after a 60 J/sq cm light dose to photosensitized cells, so it would appear that these cells are not readily affected by photodynamic effects. There are many tests for apoptosis, one of which is activation of caspases and appearance of apoptotic morphology. The best test for viability is a clonogenic assay, assuming that the cells are in an environment where they can divide. 

The pertinent question is whether the MSC population will be affected by the PDT dose used to treat glioma. If a very high light dose is required for these effects to be produced, this phenomenon is unlikely to occur in during the treatment of glioma with ALA-based PDT. The critical element is not whether ALA might have effects in the dark, but whether the light dose used was excessive for an vitro study. 
